# OpenReview forum: "Re-contextualization Mitigates Specification Gaming without Modifying the Specification"
_ICLR.cc/2026/Conference — Submitted to ICLR 2026_

### Official Review · Reviewer_HSsn · 2025-10-30

**Soundness:** 2
**Presentation:** 2
**Contribution:** 2
**Rating:** 4
**Confidence:** 4

**Summary:**

This paper aims to prevent LLMs from learning misaligned behaviors when clear supervision, like reward signals, is unavailable. The idea is pretty simple: by using different prompt templates during the inference and training procedures, the trick is named recontextualization. Experiments on three datasets demonstrate the effectiveness of the proposed method.

**Strengths:**

- The idea is fairly simple and straightforward.
- The proposed method can be used for any modern LLMs without assumptions on the architecture.

**Weaknesses:**

- About terminology:
  - I get lost since the three terminologies keep being used interchangeably: specification gaming, reward hacking, and misaligned behaviors. Would you mind further explaining their connections and differences?
- About recontextualization:
  - Just to make sure, do we need to know the potential problems ahead of time to conduct recontextualization, or can we simply use a general prompt like lines 192-193? It is only applicable to Sec. 4.1 or all datasets in Sec. 4?
  - Strictly speaking, upon recontextualization, you are not conducting on-policy learning, and you should use importance sampling to conduct non-biased estimation of the policy gradient.
  - Recontextualization seems very similar to concept erasing, and thus, the generalization is a big concern, since you can only erase one concept at a time, and all experiments in Sec. 4 are in-distribution generalization.
  - The prompt changing design is quite similar to [1].

- About experiments:

  - Just to make sure that you claim to conduct on-policy learning with GPT-4.1-mini, so you mean you are fine-tuning a GPT-4.1-mini model?
  - In Sec. 4, you choose to verify the effectiveness of the proposed method in three datasets. What is the design motivation? What are the features and differences of the three datasets?

[1] Chen, Kai, et al. "Gaining wisdom from setbacks: Aligning large language models via mistake analysis." *arXiv preprint arXiv:2310.10477* (2023).

**Questions:**

In line 263, what do you mean by "The first test case of the training test set is always incorrect"? Will the datasets collect wrong test cases for each question?

---

> ### Author Response · Authors · 2025-11-21
> **Rebuttal by Authors**
>
> We thank the reviewer for their very thoughtful questions and for highlighting areas where our explanations could be clearer!
>
> ## W1: Terminology
>
> Specification gaming occurs when models learn misbehaviors (or misaligned behaviors) that are reinforced by the reward/training signal while violating the designer’s actual intent. Specification gaming is synonymous with reward hacking, and we use “hack” synonymously with “game”. We initially mentioned this synonymity in Related Work, but have now updated the Introduction to improve clarity. Thanks for pointing it out!
>
> ## W2: Do we need to know the potential problems ahead of time to conduct recontextualization, or can we simply use a general prompt like lines 192-193?
>
> In two settings (Section 4.1, 4.3), we found that general prompts were effective. However, due to cost constraints, in this work we did not conduct a more thorough analysis of the range of prompts which are effective in each setting. We mention it in the Limitations section and leave it to future work to conduct a more in-depth analysis of what makes a prompt most effective.
>
> ## W3: Importance Sampling
>
> Thank you for highlighting this approach! Our Lie Detector GRPO experiment (the only policy gradient approach we include in the paper) does include importance sampling as part of the standard GRPO loss. However, the importance sampling is rather mild, as importance ratios are clipped to between 0.8 and 1.2.
>
> We have updated the limitations section with a discussion about importance sampling. On a theoretical level, we expect this would degrade recontextualization's effectiveness, because the prompt contrast that creates off-policiness appears central to how the method works. Importance sampling would reweight gradients by the ratio $\frac{\pi\_\theta(y|x_\text{train})}{\pi\_\theta(y|x_\text{gen})}$, effectively making updates "as if" we had generated data with the training prompt in the first place. This would approximate standard training with permissive prompts (i.e. generating data and training with a prompt encouraging misbehavior), which we found ineffective in multiple settings (Sections 4.1, 4.3). However, it is possible that some forms of importance sampling could retain the benefits of recontextualization while mitigating any adverse side effects, such as instruction-following degradation. We would be excited for future work to investigate this possibility.
>
> To that end, we conducted a small experiment to test a more aggressive form of importance sampling than occurs in our normal GRPO loss. We use vanilla importance-weighted policy gradient, scaling advantages by importance weights. For Honest -> Lie recontextualization, we found that standard GRPO mitigated deception more effectively and achieved the higher ground truth reward (aligning with the theoretical explanation above). However, these results represent one possible implementation of importance sampling and thus do not provide conclusive evidence on whether importance sampling is worthwhile for recontextualization. We include these results in Appendix E.9, and now explicitly encourage further investigation of important sampling in our Limitations section.
>
> The following resultls are after 60 steps of training. We find that using the standard GRPO algorithm mitigates deception more effectively while achieving the same training reward (see Appendix E.9 for full details).
>
> | Method applied to Honest -> Lie Recontextualization | Deception Rate (%) | Training Reward | Ground Truth Reward |
> | --------------------------------------------------- | ------------------ | --------------- | ------------------- |
> | Standard GRPO                                         | **8.67 ± 1.45**    | **0.99 ± 0.01** | **0.87 ± 0.02**     |
> | Importance Sample                                   | 26.00 ± 7.00       | **0.99 ± 0.02** | 0.67 ± 0.06         |

---

> > ### Author Response · Authors · 2025-11-21
> > **Rebuttal by Authors, Continued**
> >
> > ## W4: Concept erasure
> >
> > We agree this is an important concern, and our approach does share some similarity with concept erasure: within a given training context, the policy may effectively “lose” the ability to execute a particular reward-hacking strategy. However, our goal is not to remove an underlying concept from the model’s representation, but to guide the policy away from failure modes in the RL environment so that it learns the intended task. Rather than trying to entirely delete an internal concept that might reappear elsewhere, we focus on preventing the problematic behavior from arising during learning, which is especially relevant since reward hacking can naturally extend to more broadly misaligned behaviors. Thus, it is sufficient to prevent such concepts from being used to optimize reward in the training environment. Finally, while concept-erasure methods are often framed as operating on one concept at a time, our setup offers some flexibility: different prompts can target different or even multiple behaviors, and our results in sections 4.1 and 4.3 shows that a general prompt can work better than over-specific ones.
> >
> > ## W5: Wisdom from setback
> >
> > Thank you for mentioning that work. It was an interesting read and the prompt changing strategy indeed bears similarities with our approach. We think it is a relevant source and have added it to our related work section.
> >
> > ## Experiment Clarifications:
> > 1. Regarding whether our initial experiment uses data from GPT-4.1-mini:
> >
> > Yes, the data is generated from GPT-4.1-mini, and thus on-policy. We realize that the section was ambiguous and we’ve clarified it. There was additionally a typo in our example prompt-completion pair stating that the completion came from GPT-4o, which we’ve fixed. Thank you for raising the clarifying question!
> >
> > 2. Why did we use these 3 datasets?
> >
> > We selected datasets that we felt mirror real-world alignment challenges at a manageable scale. All of our 4 experimental settings (we’ve added a new one in Section 4.4) reinforce some undesired behavior, or “hack”. 3 of these 4 settings additionally reinforce a beneficial capability we want to develop: writing correct code (Section 4.2), achieving high training reward (Section 4.3), and producing high-quality assistant responses (Section 4.4). The exception is the evaluation gaming setting (Section 4.1), which only includes quality verification of trained model outputs (Appendix C.4).
> >
> > These environments address distinct alignment concerns. First, models that prioritize evaluation metrics over soundness exemplify a general risk of Goodhart's law in RL. Second, coding models may optimize for passing tests or executable code at the expense of long-term user satisfaction (METR, 2025b). Third, post-training processes could inadvertently reinforce deceptive behavior (Casper et al., 2023; OpenAI, 2023b). We also included the SoLiD lie detector environment specifically to connect with existing literature on these issues.
> > Despite training a desired capability, the coding and lie detector settings (Sections 4.2 and 4.3) are relatively narrow. The coding environment uses a restricted format of single function python coding challenges, while the lie detector casts the model as a corporate assistant responding to similarly structured single-turn prompts.
> >
> > To address this limitation, we added a fourth setting (Section 4.4) that simulates an RLHF reward model by simultaneously reinforcing both coherent, high-quality assistant responses and sycophancy. This setting uses a more diverse training dataset and entrains a more challenging capability for a base model: responding competently across varied assistant interactions.
> >
> > ## Q1:  Regarding "the first test case of the training test set is always incorrect"
> >
> > In this setting, the model receives a coding problem together with test cases that will be used to evaluate its solution. These test cases include an input and output value. For the first test case, the output value is incorrect. An example can be found in Table 8. We adapted the phrasing to remove the potentially confusing reference to the training set.
> >
> > We thank the reviewer once again for their feedback and thorough engagement with our work!

---

> ### Comment · Reviewer_HSsn · 2025-11-24
>
> I have read the authors' rebuttal, which addresses most of my initial concerns. I can see the effectiveness of the proposed method, but the authors should clearly clarify the method design with precise terminology, formulations, and potential discussions. I'm happy to see the authors complement the missing parts in the revised version, especially the importance sampling experiments in Appendix E.9. I wish my comments could help the authors better revise their paper. I have updated my score :)

---

### Official Review · Reviewer_v1Zs · 2025-10-31

**Soundness:** 3
**Presentation:** 3
**Contribution:** 2
**Rating:** 4
**Confidence:** 2

**Summary:**

This paper proposes recontextualization, a lightweight modification to on-policy fine-tuning that mitigates specification gaming—cases where models exploit imperfect rewards. The method generates responses under safe prompts (e.g., “Be honest”) but trains as if they were responses to permissive prompts (e.g., “Lie to the user”). Experiments on evaluation metric gaming, code test hacking, and deception show reduced misbehavior without modifying the reward model or adding data. The approach is simple, broadly applicable, and interpretable as a form of context distillation for alignment.

**Strengths:**

Novel conceptual framing: The paper reframes misalignment mitigation as a contextual shift problem, introducing a lightweight intervention that requires only prompt-level changes rather than new supervision.

Clear motivation and connection to prior work: Builds coherently on context distillation, HER, and alignment research (Constitutional AI, RLHF pathology studies).

Empirical breadth: Evaluated across three qualitatively distinct gaming behaviors (reward metric overfitting, test hacking, deception), showing consistent mitigation.

Strong intuition and simplicity: Implementation is easy to integrate into existing RLHF or GRPO pipelines. The clarity of the A→B notation and controlled experiments is commendable.

**Weaknesses:**

All experiments occur in small, highly controlled settings (e.g., GPT-4.1-mini, MBPP with injected faulty test cases, DolusChat deception toy dataset). None convincingly demonstrate that the method scales to real RLHF or multi-turn alignment pipelines. This limits external validity.

The paper admits that recontextualization makes the data off-policy, which can bias gradient estimates in on-policy algorithms like GRPO. There is no ablation quantifying the magnitude of this mismatch. It is possible that the observed gains stem from mild regularization rather than the proposed mechanism.

The baselines are relatively weak—e.g., “stronger KL regularization”—and do not include more sophisticated mitigations (e.g., regularized PPO, debate-based oversight, counterfactual data augmentation). Without these, claims of competitiveness might be overstated.

**Questions:**

Does recontextualization still help when the reward signal is not misspecified (i.e., no obvious gaming behavior)?

How would recontextualization behave in long-horizon multi-turn RLHF (e.g., dialogue agents with memory)?

---

> ### Author Response · Authors · 2025-11-21
> **Rebuttal by Authors**
>
> We thank the reviewer for their very thoughtful comments! We appreciate your comments on the clarity and novelty of our framing, as well as the clear motivation and empirical breadth of our work.
>
> ## W1: Experiments occur in small, highly-controlled settings
>
> We agree that this has been a limitation of our work. To that end, we have tested recontextualization in an additional more realistic setting and added this experiment to the uploaded paper (Section 4.4). Our setting trains an open-source base model, via expert iteration, to become a more competent assistant. Yet, our training signal reinforces both quality and sycophancy in assistant responses (as do real-world reward models), and we find recontextualization achieves quality on par with standard training while mitigating the learning of sycophancy. We believe this setting is more realistic in two respects:
>
> 1. Producing high-quality assistant responses (for the pretrained model) is a more demanding task than occurred in our existing settings.
> 2. The data mixture is more varied. Whereas our prior datasets were quite narrow, our training dataset for this setting includes 7200 prompts, half of which are generic helpfulness data, and half of which give the model a clear opportunity to be sycophantic.
>
> Due to compute and resource constraints, this setting is still limited in its realism, lacking the scale of real world post-training and a production reward model (which we simulate via an LLM judge). However, we believe it improves upon the realism of our prior experiments and provides additional evidence for recontextualization’s potential in full production environments. We are excited for future work which evaluates this method in such settings, and hope that our paper and the source code released alongside it will aid such efforts.
>
> ## W2: No ablation quantifying the magnitude of the off-policiness. Effect might stem from mild regularization.
>
> We agree that mild regularization is an important explanation to rule out. We believe our paper already includes 3 results that confirm the observed gains do not stem from mild regularization, although we did not explicitly call them ablations.
>
> 1. If the effect were from mild regularization, semantically-unrelated recontextualization prompts would be effective. We confirm that they are not in the Evaluation Gaming experiment (the "Control" prompts in Figure 3, Section 4.1). The fact that we see no deviation at all from standard training for any of these prompts implies that minor regularization is not impacting training.
>
> 2. If the effect were from mild regularization, generating from Exploit and training with Don't-Exploit (which similarly takes the model off-policy) would be effective. Instead, it **substantially boosts** evaluation gaming relative to standard training, confirming that the semantic contrast between prompts drives the effect.
>
> 3. If recontextualization were merely limiting generalization from 'Misbehave' training contexts to 'Neutral' evaluation contexts, we would see that generating and training from the 'Misbehave' context would be effective (shown to be ineffective in Section 4.1 and 4.3). We’d also see misbehavior emerge when evaluating on the 'Misbehave' context, which we do not (see Appendix B).
>
> To address your point about quantifying off-policiness, we've updated our paper to track the magnitude of "off-policiness" throughout GRPO training (Appendix E.3). We operationalized "off-policiness" as the KL divergence between the model's distributions conditioned on the generation vs. training context. We find sharp initial declines for the first ~40 training steps, and then the magnitude of the off-policiness plateaus.

---

> > ### Author Response · Authors · 2025-11-21
> > **Rebuttal by Authors, Continued**
> >
> > ## W3: Baselines are Relatively Weak, e.g. "Stronger KL regularization"
> >
> > We thank the reviewer for highlighting this - we have now added additional baselines for our lie detector setting (Appendix E.2). We regularize on OOD alignment data (harmlessness data and honesty data), which has been shown to be effective at preventing misaligned generalization [1] [4]. However, our setup is somewhat distinct from those this technique was validated on; we're not trying to prevent far-reaching OOD misalignment, but rather in-distribution misalignment. Nevertheless, we believe these are valuable baselines to run. They may be considered counterfactual data augmentation [2], which the reviewer suggested—the included honesty and harmlessness data opposes the bias to the training reward.
> >
> > We also believe that our initial KL regularization baseline (increasing the weight on the standard regularization penalty within the GRPO loss) is a fairly strong baseline for this setting. The weight on this penalty was found by the authors of SoLiD [3] to be a critical factor for whether the model learned deception. The other factors they discovered were a) recall of the lie detector and b) RL algorithm used, e.g. GRPO, both of which are assumed to be fixed in our setup. We've updated the paper to reflect this context.
> >
> > Regarding your suggested baselines:
> > 1. Regularized PPO: We performed regularized GRPO, which includes a KL penalty on training data and also implements PPO-style clipping.
> > 2. Debate-based oversight: Debate is an interesting research agenda for scalable oversight. We consider it to be out of scope for this experiment, as it requires substantial design and engineering effort, and we have compared to 3 baselines that are similarly lightweight to recontextualization.
> >
> > [1]: Kaczér et al. “In-Training Defenses against Emergent Misalignment in Language Models” (2025)
> > [2]: Lu et al. “Gender Bias in Neural Natural Language Processing” (2019)
> > [3]: Cundy et al. “Preference Learning with Lie Detectors can Induce Honesty or Evasion” (2025)
> > [4]: Azarbal et al. "Selective Generalization: Improving Capabilities while Maintaining Alignment" (2025)
> >
> > ## Q1: Does recontextualization still help when the reward signal is not misspecified?
> >
> > This is an interesting question! In our initial submission, we compared recontextualized vs. standard approaches when training on a **strong** lie detector (Appendix E.6). Recontextualization performs similarly to standard training, although slightly worse in terms of ground truth reward. Recontextualized training reduces deception to ~6%, and standard training reduces deception to ~4% (from a pre-GRPO rate of ~24%).
> >
> > To further address your question, we added an experiment testing a correctly specified reward signal in the code setting (Section 4.2). Completions are scored based on the correct version of the test cases. With a correct training signal, we observe that recontextualization further increases the rate of correct answers. We have included these results in Appendix D.6.
> >
> > ## Q2: How would recontextualization behave in long-horizon multi-turn RL?
> >
> > This remains an open question, and we view long-horizon, multi-turn settings as follow-up work rather than something we fully address here. In principle, one could extend our approach by recontextualizing at the level of the full dialogue by modifying the initial system prompt, by editing the user/assistant chat history, or even by recontextualizing at each turn before the next generation. We leave the design and evaluation of these more complex recontextualization approaches to future work.
> >
> > We thank the reviewer once again for their very thoughtful feedback and suggestions for improvement!

---

### Official Review · Reviewer_ooJC · 2025-11-01

**Soundness:** 2
**Presentation:** 3
**Contribution:** 2
**Rating:** 2
**Confidence:** 3

**Summary:**

The current work puts forth the concept of recontextualization, which the authors term as a reduction in how often LLMs are able to "game" or subvert training signals. In particular, they demonstrate how such recontextualization stops models from learning misbehaviors such as reward hacking on evaluation metrics at the expense of overall text generation quality.

**Strengths:**

1. the background information regarding specification gaming is well presented and necessary to understand the motivation behind the current work.
2. the motivation for why recontextualization provides an avenue for weak supervision is documented well in the discussion.

**Weaknesses:**

1. use of GPT-4o in an LLM-as-judge paradigm. There have been many works which have demonstrated the self-preference bias of these models (e.g., https://aclanthology.org/2024.acl-long.511/, https://arxiv.org/abs/2404.13076, https://aclanthology.org/2024.acl-long.826/), so it would have been more helpful to utilize multiple judges, or at least more than one judge in order to ensure reproducibility and a lack of judge-related bias that could potentially impact downstream results.
2. use of only GPT-4.1-mini in experiments. despite this being a performant model, it is a closed source model and thus the reproducibility of the experiments is brought into question. Why not also use an open-weights model in addition?

**Questions:**

N/A, see above.

---

> ### Author Response · Authors · 2025-11-21
> **Rebuttal by Authors**
>
> We thank the reviewer for their comments, and for highlighting the clarity of the motivation for this work. In particular, we want our work to be fully reproducible and appreciate the reviewer pushing for a high degree of reproducibility. We hope to address the mentioned concerns.
>
> ## W1: Use of GPT-4o in an LLM-as-a-judge paradigm
>
> In our experiments, we use GPT-4o as a judge to evaluate generations from GPT4.1-mini (Section 4.1) and Llama-3.1-8B-Instruct (Section 4.3). Moreover, we never ask our judge to rank responses between different models, the scenario in which self-preference bias would typically emerge. Therefore, we do not believe our experiments are vulnerable to self-preference bias of the LLM judge.
>
> ## W2: Use of only GPT-4.1-mini in experiments
>
> In addition to GPT4.1-mini, our original submission also used the open-weight Llama-3.1-8B-Instruct for the lie detector experiment in Section 4.3. As part of this rebuttal, we have added an additional more realistic setting in Section 4.4, and for that experiment we used a different open-weight model (Qwen3-8B-Base) to provide further evidence that this method works across model types.
>
> We ensure the reproducibility of all our experiments by releasing our codebase as supplementary material, allowing others to directly rerun our pipelines. For those interested in using open-weights models in the first two experimental settings, the data-generation and fine-tuning modules can be readily adapted and integrated into the main workflow using the framework of their choice.
>
> ## Request for additional feedback
>
> We believe we've addressed the mentioned weaknesses in your review. However, we notice that you have substantial concerns about the work, as reflected in your score. We'd like to ask if you have additional weaknesses in mind that went unstated in your initial review. We would welcome additional feedback and the opportunity to address it.
>
> Thank you again for your time and feedback thus far.

---

> > ### Comment · Reviewer_ooJC · 2025-11-26
> >
> > Thanks to the authors for their reply. I will maintain my score.

---

### Official Review · Reviewer_iB2c · 2025-11-01

**Soundness:** 3
**Presentation:** 4
**Contribution:** 3
**Rating:** 8
**Confidence:** 4

**Summary:**

This paper introduces recontextualized training. In recontextualized training, you sample model completions given one system prompt and then train the model with a different system prompt. Specifically, you sample using a system prompt which says that taking a particular action the AI might be incentivized to do is bad and then train it with a system prompt that says that that action is fine. The authors find that re-contextualization reduces the extent to which training on these examples of bad behavior generalizes to the model doing other bad behavior, while maintaining performance.

**Strengths:**

This is a valuable contribution on an important topic. This algorithm is a pretty smart idea and I think that the results are pretty compelling. The results are definitely compelling enough that this technique deserves further study.

**Weaknesses:**

I think the settings studied in this paper are sort of toy. This makes me worry that on more complicated settings where RL is harder, the technique would fail for a reason that these settings were too easy to demonstrate. It would be great to study some more realistic settings. Experiments in more realistic settings would make it much easier to understand whether there are caveats associated with this technique, or difficulties with adopting it in practice. I strongly suspect such caveats exist, and it would be good to know about them.

It would also be nice to know what happens with larger models, of course.

I share the author's concern that the off-policy nature of the training sequences here is kind of scary. I don't know RL well enough to know how likely this is to cause huge issues. All the examples of off-policy trading strategies that the authors mention (e.g. hindsight experience replay) are evidence that this isn't a big problem.

**Questions:**

Maybe the easiest way for you to make this paper stronger from my perspective would be to speculate about how to handle the problems you mention about these updates being off policy. My understanding is that the RL community has handled issues like this in the past, but I don't know that literature well enough to know whether this kind of technique is likely to cause huge issues, or whether those issues are very likely to be resolvable.

I'm slightly concerned that for emergent misalignment style reasons, training on models that have been told to behave badly will cause the model to learn to behave badly. Are you worried about this?

---

> ### Author Response · Authors · 2025-11-21
> **Rebuttal by Authors**
>
> We thank the reviewer for their very thoughtful review and for highlighting the value of the contribution.
>
> ## W1: More realistic setting
>
> We agree that this has been a limitation of our work. To that end, we have tested recontextualization in an additional more realistic setting and added this experiment to the uploaded paper (Section 4.4). Our setting trains an open-source base model, via expert iteration, to become a more competent assistant. Yet, our training signal reinforces both quality and sycophancy in assistant responses (as do real-world reward models), and we find recontextualization achieves quality on par with standard training while mitigating the learning of sycophancy. We believe this setting is more realistic in two respects:
> 1. Producing high-quality assistant responses (for the pretrained model) is a more demanding task than occurred in our existing settings.
> 2. The data mixture is more varied. Whereas our prior datasets were quite narrow, our training dataset for this setting includes 7200 prompts, half of which are generic helpfulness data, and half of which give the model a clear opportunity to be sycophantic.
>
> Due to compute and resource constraints, this setting is still limited in its realism, lacking the scale of real world post-training and a production reward model (which we simulate via an LLM judge). However, we believe it improves upon the realism of our prior experiments and provides additional evidence for recontextualization’s potential in full production environments. We are excited for future work which evaluates this method in such settings, and hope that our paper and the source code released alongside it will aid such efforts.
> ## W2: Larger models
> We agree that larger models would provide additional evidence for the validity of the technique in production settings. Our source code is available, and we strongly support future work on models larger than GPT4.1-mini (for Sections 4.1 and 4.2), and open-source models larger than 8B parameters (for Sections 4.3 and 4.4).
> ## Q1: Off-policy nature of training
> We agree that further speculation and investigation on this issue is valuable! We have updated the paper in a few respects to address this:
>
> 1. Investigating coherence after recontextualization
>
> A concern we've had is that the off-policy nature of recontextualization could impact models' general coherence. We conducted investigations into recontextualized GPT-4.1-mini models' MMLU and instruction-following (IFEval) scores. While we found no reduction in MMLU scores, we did find a minor decline in instruction-following (Appendix C.5). This is also true for recontextualized Llama-3.1-8B-IT in the Lie Detector setting (Appendix E.3).
>
> 2. Mitigating adverse effects on instruction following
>
> In Appendix E.4, we find we can mitigate a majority of the decline in instruction-following for Llama-3.1-8B-IT by adding a KL regularization term on OOD chat data during GRPO, without affecting in-distribution performance.
>
> 3. Testing importance sampling
>
> We have updated the limitations section with a discussion about importance sampling as a way to correct for the distribution shift between generation and training prompts. However, we expect this would degrade recontextualization's effectiveness, because the prompt contrast that creates this distribution shift appears central to how the method works. Importance sampling would reweight gradients by the ratio $\frac{\pi\_\theta(y|x_\text{train})}{\pi\_\theta(y|x_\text{gen})}$, effectively making updates "as if" we had generated data with the training prompt in the first place. This would approximate standard training with permissive prompts (i.e. generating data and training with a prompt encouraging misbehavior), which we found ineffective in multiple settings (Sections 4.1, 4.3). However, it is possible that some forms of importance sampling could retain the benefits of recontextualization while mitigating any adverse side effects, such as instruction-following degradation. We would be excited for future work to investigate this possibility.
>
> ## Q2: Emergent Misalignment concerns
>
> This is certainly an intuitive concern. However, we are not particularly worried. A model becoming misaligned during training depends primarily on how aligned its completions are *relative to its prior for the context*. This was illustrated in our Evaluation Gaming (Section 4.1), where we found that generating data and training with *Exploit* prompts led to approximately the same results as generating and training with *Neutral* prompts. Even though the model's completions included worse behavior when generating/training with *Exploit*, the completions were not worse *than expected* for that context. Recontextualization leverages this effect, and we show that training the model with bad instructions is counterintuitively beneficial when the generations come from a better distribution prior.
>
> We thank the reviewer once again for their thoughtful feedback!

---

### Author Response · Authors · 2025-12-02
**Author Remarks on Paper Improvements**

We thank the reviewers for their constructive comments.

We are glad to see our work received as *“a valuable contribution on an important topic,”* with a *"novel conceptual framing"*. Our algorithm for mitigating specification gaming has *“strong intuition and simplicity”* and achieves *“results [that] are pretty compelling” “across three qualitatively distinct gaming behaviors”*.

## Experimental improvements during rebuttals

1. **Adding a more realistic experimental setting**

Reviewers iB2c and v1Zs suggested we further validate recontextualization in a more realistic environment. We *added an entirely new experimental setting to our paper (Section 4.4)*. This setting includes a more realistic data composition and a more demanding learning task, modeling the development of sycophancy from real-world post-training.

2. **Importance Sampling**

Reviewer HSsn requested we use importance sampling during recontextualization. We've now tested this approach in Appendix E9 and found it to be less effective, aligning with our theoretical predictions (explained in our rebuttal and added to the paper’s Discussion section and Appendix E9). Reviewer HSsn was satisfied by this addition.

3. **Additional baselines for Section 4.3**

Reviewer v1Zs highlighted limited baselines in our GRPO experiment. We added 2 additional baselines in Appendix E2.

4. **Quantifying off-policiness**

Reviewer v1Zs requested we quantify off-policiness, which we have now included in Appendix E3.

4. **Experiment with correctly specified training signal**

Reviewer v1Zs asked: “does recontextualization still help when the reward signal is not misspecified?”. Our original submission included one experiment addressing this question in Appendix E6. We added another experiment addressing this question in Appendix D6.


## Other comments addressed

1. **Clarifying already-present ablations.** Reviewer v1Zs raised concerns about the lack of ablations confirming that recontextualization works due to prompt semantics. In fact, our paper includes 3 such ablations, though they were not explicitly labeled as such, which may have contributed to the oversight. We discuss these in detail in our response to Reviewer v1Zs.

2. **Terminology clarifications.** We improved our work’s presentation and terminology following reviewer HSsn’s questions.

We thank the reviewers again for the valuable discussions, and the Area Chair for their consideration.

---

### Meta-Review · Area_Chair_rt3d · 2026-01-07

**Summary:**

Summary of main concerns:

1. Experiments are not realistic. (iB2c, v1Zs, HSsn)
2. Data is off-policy, which could lead to issues in training. (iB2c, v1Zs, HSsn)
3. Baselines are weak. (v1Zs)

**Reviewer Concerns:**

I think (3) is mostly addressed since the authors added new baselines. (2) still requires more analysis and a larger discussion. For (1), the authors added new experiments with the Anthropic HH dataset, but the dataset is still highly targeted, and the evaluation is still largely in-distribution. The authors did show results on MMLU and IFEval in the Appendix, but these are not fully convincing because they are somewhat mixed, and a more thorough investigation is needed.

**Reviewer Scores:**

I expect reviewers HSsn and v1Zs would have increased their scores. I expect the other two scores would have remained the same.

---

### Decision · Program_Chairs · 2026-01-26

Reject